# Railway Wheel Flat Detection System Based on a Parallelogram Mechanism

**DOI:** 10.3390/s19163614

**Published:** 2019-08-20

**Authors:** Run Gao, Qixin He, Qibo Feng

**Affiliations:** MoE Key Lab of Luminescence and Optical Information, Beijing Jiaotong University, No. 3 Shangyuancun, Beijing 100044, China

**Keywords:** wheel flat detection, multibody dynamic theory, parallelogram mechanism

## Abstract

Wheel flats are a key fault in railway systems, which can bring great harm to vehicle operation safety. At present, most wheel flat detection methods use qualitative detection and do not meet practical demands. In this paper, we used a railway wheel flat measurement method based on a parallelogram mechanism to detect wheel flats dynamically and quantitatively. Based on our experiments, we found that system performance was influenced by the train speed. When the train speed was higher than a certain threshold, the wheel impact force would cause vibration of the measuring mechanism and affect the detection accuracy. Since the measuring system was installed at the on-site entrance of the train garage, to meet the speed requirement, a three-dimensional simulation model was established, which was based on the rigid-flexible coupled multibody dynamics theory. The speed threshold of the measuring mechanism increased by the reasonable selection of the damping coefficients of the hydraulic damper, the measuring positions, and the downward displacements of the measuring ruler. Finally, we applied the selected model parameters to the parallelogram mechanism, where field measurements showed that the experimental results were consistent with the simulation results.

## 1. Introduction

Wheels are the most important bearing and moving parts of a train that have a crucial impact on driving safety. Railway wheel flats are induced by unintentional sliding between the wheel and rail at braking [1,2]. The additional periodic impact load caused by wheel flats is related to the depth and length of flats, as well as the train speed and load, which even can be several times the wheel static load [3,4,5,6]. Wheel flats are the main causes of wheel bearing damage, axle temperature rise, axle fracture, as well as rail and concrete sleeper fracture [7,8]. Rapid and accurate detection of wheel flats is important to ensure the train operation safety.

At present, the commonly used on-line detection methods for wheel flats include the ultrasonic method [9,10,11], wheel impact load detection method [12,13,14,15,16,17,18,19,20,21], parallelogram method [22], and so on. The ultrasonic method is divided into the electromagnetic ultrasonic method [9] and the ultrasonic ranging method [10,11]. Salzburger et al. [9] used an electromagnetic ultrasound to measure wheel flats. When the wheel passes through, the electromagnetic ultrasonic probe embedded in the rail surface contacts the wheel tread, and the probe will stimulate the ultrasonic wave to spread inside the wheel. When a flat exists on a wheel tread, the echo will be generated, and the tread information can be obtained by analyzing the echo signal. This method is suitable for low-speed measurement below 15 km/h, and it is convenient for measurement. However, the original structure of the rail will be damaged during installation, and the wheel flat cannot be quantitatively detected. Brizuela et al. [10] used ultrasound pulses (Rayleigh waves) to detect wheel flats by detecting the variations in the round-trip time of the ultrasound pulse to the rail–wheel contact point. The insufficiency of this method was that the measuring accuracy was affected by the measuring distance, so it was difficult to accurately detect the wheel flat of the whole circumference of the rolling wheel. The wheel impact load detection method is used to detect the impact force of a wheel flat on the rail by installing a series of strain gauge [12], accelerometer [13,14,15,16,17], optical fiber sensor [18,19,20], piezoelectric cable [21], and other sensors at different positions along the rail. Stratman et al. [12] installed a large number of strain gauges on the track to measure the impact force caused by wheel flats, and they indirectly detected the tread abrasion level. Belotti et al. [13] used an accelerometer to measure vibration impact force, and wheel flats at different speeds were quantized by wavelet transform. Filograno et al. [18] used a fiber Bragg grating (FBG) sensor to detect wheel-rail force at a train speed up to 350 km/h. In their experiments, the data were analyzed in a time domain and frequency spectrum to detect wheel flats and roundness. Bracciali et al. [21] proposed a low-cost wheel flats detector (WFD). The WFD was a special clamp with piezoelectric cable, which was stuck onto the bottom of the rail to detect the sudden signal of wheel tread flats.

Among the above-mentioned detection methods, the wheel impact detection method is widely used. The wheel impact load method has a wide application range, simple structure, and low technical difficulty, but it is limited to qualitative measurement. To quantify the wheel flat, some researchers use the methods of calibration and time-domain analysis to quantify the relationship between wheel flat length and impact force. However, the measurement results are easily affected by several external factors, such as train speed, train weight, railway tracks, rail quality, and electromagnetic interference, making it difficult to measure with high accuracy [18,22]. Therefore, the quantitative measurement of wheel flats has become a key problem in wheel tread flat detection. In this paper, a method using the parallelogram mechanism was proposed to detect wheel flats dynamically and quantitatively. The wheel flat detection system was built with high sensitivity and high interference resistance. The parallelogram mechanism consists of the measuring ruler, an eddy current sensor, a hydraulic damper, and so on. The wheel flat can be detected by measuring the vertical displacement change of the measuring ruler. To improve the system performance, a three-dimensional simulation model was established based on the rigid-flexible coupled multibody dynamics theory and the measuring ruler in the model was made flexible using the finite element method. The parameters of the system were reasonably selected via simulation experiments of the damping coefficients of the hydraulic damper, the measuring positions, and the downward displacements of the measuring ruler. The model was verified by field tests and presented theoretical evidence for system installation and selection principle of components.

## 2. Measuring Mechanism

The parallelogram mechanism was used to detect the wheel flat depth dynamically and quantitatively using the vertical displacement change of the flat wheel. This section mainly introduces the theory and senor structure.

### 2.1. Theory

The motion diagram of a wheel flat is shown in Figure 1a. Point O in the figure represents the center of wheel circle, v represents the running speed. According to the geometric structure of the circle, the wheel flat length *L* can be calculated as Equation [10]:(1)$$L\;=\;2\sqrt{{2Rd\;-\;d}^{2}}\;\approx \;\sqrt{8Rd}$$
where d is the maximum wheel flat depth and *R* is the wheel radius. As Figure 1a shows, as point *B* is rolling and falling to the ground, the wheel center (point *O*) moves in circles around the end of the wheel flat (point *A*), reaching point *O*′. In this process, the distance between point *O* and point *A* is R, the rotation speed of point *O* is ω = ν/R, and the motion equation of point *O* in the vertical is as follows:(2)y(t) = R[cos(ωt) − 1]
where $0\;<\;t\;<\;(R/v)\arcsin(\frac{L}{2d})$. While the flat wheel is rolling, the vertical displacement response curve of point *O* is as shown in Figure 1b. As Figure shows, the maximum vertical displacement of point *O* represents the wheel flat depth d, and the wheel flat length L can be calculated according to formula (1).

### 2.2. Sensor Structure

The parallelogram mechanism is the core component of the wheel flats dynamic measurement system, which is mainly composed of the measuring ruler, connecting rods, springs, hydraulic damper, limit block, and eddy current sensor, as shown in Figure 2. The impact force produced by the collision between the wheel and measuring mechanism was relieved by a hydraulic damper (SMC Corporation, Tokyo, Japan, model RB-ADA505MPC). An eddy current sensor (Hunan TianRui Corporation, Hunan, China, model trin03-28) was used to detect the displacement of the measuring ruler. The measuring range of the eddy current sensor was from 3.75 mm to 19.73 mm, and the resolution was 35 μm.

When a train passes through the measuring device, the wheel presses down the upper plate (called measuring ruler) of the parallelogram mechanism, and the vertical displacement between the sensor and the measuring ruler is measured by the eddy current sensor. Under ideal conditions (i.e., the parallelogram mechanism has no machining error and is in an ideal assembly position), the output waveforms of the standard wheel and the worn wheel are as shown in Figure 3. If the measured wheel is a standard wheel, then the vertical displacement of the detection signal will not change. If the flat exists on the measured wheel tread, a downward displacement change will occur in the detection signal at the flat position, and the vertical motion process of the flat wheel is as shown in Figure 1b. Given the contact between the parallelogram mechanism and the wheel, the downward displacement signal can be generated by the eddy current sensor, and the maximum vertical displacement represents the wheel flat depth. The wheel flat length can be further calculated according to Formula (1). In addition, the wear depth of the worn wheel can be obtained by comparing it with the standard wheel.

## 3. Model Establishment

To improve system performance, the ADAMS dynamics simulation software was employed to establish the rigid-flexible coupled three-dimensional model of the measuring mechanism, as shown in Figure 4. Dynamics were widely used to analyze the complex motion relations of objects. Consequently, this section does not introduce the theory of dynamics, but it introduces modeling [23,24,25]. In the model, the wheel, the rail, and the parallelogram mechanism, except the measuring ruler, were set as the rigid bodies. We found in the actual measurement that the measuring ruler was flexibly deformed under the external force. Therefore, the measuring ruler was simulated as a flexible body using the finite element technique. The measuring ruler was hinged to the connecting rods, and springs and hydraulic dampers were set as flexible connectors. The parallelogram mechanism was made of steel, with the following parameters: Young’s modulus *E* = 207 GP, Poisson’s ratio ν = 0.29, and density ρ = 7801 kg/m^3^. The contact force between the rigid body wheel and the flexible measuring ruler was set such that the wheel applied pressure to the parallelogram mechanism. The motion process of the measuring ruler was simulated based on the multibody dynamics method while the wheel was moving.

## 4. Parameter Selection

The three-dimensional model was used to simulate the motion process of the parallelogram mechanism under the wheel pressure. It was necessary to analyze the influencing factors of the measurement based on the simulation results and to reasonably select the parameters to improve the performance of the measuring mechanism.

### 4.1. Influencing Factors of Measurement

The influencing factors of measurement mainly included two parts: the measuring ruler deformation and impact vibration of the measuring ruler. The simulation model could be used to analyze the deformation of the measuring mechanism and compared it with the experiment. Additionally, it was necessary to analyze the motion process of the measuring ruler under different speeds to obtain the velocity threshold value of the measuring ruler vibration.

#### 4.1.1. Measuring Ruler Deformation

The flexible deformation of the measuring ruler will affect the detection accuracy of the wheel tread flat. Under ideal conditions, the measuring ruler is a rigid body where no deformation occurs under the external force, and the simulation curve of the rigid measuring ruler is shown as the red curve in Figure 5a. In actual measurements, the measuring ruler is deformed flexibly under the wheel pressure, and the deformation is related to the stress conditions. After comparing the simulation curve of the flexible measuring ruler with the experimental curve in Figure 5a, the simulation results were in good agreement with the actual measurement, but the deformation at the rear part in the experimental curve was higher than the front part by 0.8 mm. The reason for this phenomenon may be that the parallelogram mechanism was not completely parallel to the rail during installation, which resulted in different displacements at both ends of the measuring ruler. As Figure 5b shows, the greater the stiffness of the material, the smaller the deformation of the measuring ruler. Therefore, materials with large Young’s modulus *E* should be chosen.

#### 4.1.2. Impact Vibration of the Measuring Ruler

When a wheel collides with a measuring ruler at a certain speed, there are two kinds of forces: the normal impact force and the tangential friction force. The tangential friction can be calculated using Coulomb friction, and the impact function of normal impact force can be expressed as follows:(3)$$F=k\cdot \Delta x^{e}+\mathrm{STEP}(\Delta {x,0,0,d}_{max}{,c}_{max})\cdot \frac{d\Delta x}{dt}$$
where k represents the material stiffness, Δx represents the amount of extrusion deformation caused by a collision between two objects, *e* represents the index which is used to calculate the contribution value of the material stiffness term in instantaneous normal force, $d_{max}$ represents the maximum allowable penetration depth, and $c_{max}$ represents the maximum damping value applied when the maximum penetration depth is reached. STEP is a transition function. The first parameter of the STEP function represents the independent variable Δx, and the value range of Δx is (0, $d_{max}$), and the corresponding output value range of STEP is (0, $c_{max}$). According to Formula (3), the impact force is related to the velocity of impact at that moment. The higher the velocity, the greater the impact force. The wheel impact force will cause a vibration of the measuring mechanism and affect the detection accuracy.

The simulation results of the measuring ruler motion process at different speeds are shown in Figure 6a, while Figure 6b shows the relation between the maximum vibration displacement of the measuring ruler and the vehicle speed. We found that system performance was influenced by the velocity of the train. The vibration of the measuring ruler was negligible when the speed was below a certain threshold (14 km/h), and when the train speed was higher than 14 km/h, the vibration displacement of the measuring ruler rapidly increased. When the speed was 15 km/h, the vibration displacement of the measuring ruler reached a value of 0.5 mm. If the train passed through the measuring mechanism at this speed, it would be misjudged because of the vibration of the measuring ruler.

### 4.2. Parameter Selection

We conducted parameter selection simulation experiments to improve the performance of the measuring mechanism. By adjusting the damping coefficient, the measuring position, and the downward displacement of the measuring ruler, we obtained the optimal parameters, and the performance of the measuring mechanism was improved.

#### 4.2.1. Selection of the Damping Coefficient of a Hydraulic Damper

The damping coefficient of the hydraulic damper will affect the motion state of the measuring ruler. A hydraulic damper is a kind of a vibration control device that is commonly used to control impact vibration and it is sensitive to speed response. We performed simulations with different damping coefficients, and the results are shown in Figure 7. The vehicle speed was set to 14 km/h. When the damping coefficient *c* was 0.1 N·s/mm, the measuring ruler would vibrate violently; and when the *c* increased to 0.2 N·s/mm or above, the damper had a better control effect on the vibration of the measuring ruler. When the *c* was set to 20 N·s/mm, the measuring ruler could not return to the initial position after the wheel left, and the rebound displacement of the measuring ruler decreased with the increase of the damping coefficient. When *c* = 40 N·s/mm or more, the maximum deformation of the measuring ruler was close to the rebound displacement, which easily caused the missing measurement due to the loss of the benchmark of the wheel. In summary, the optimal damping coefficient of the hydraulic damper should be in the range of 0.2–20 N·s/mm.

#### 4.2.2. Selection of the Measurement Position

Different measuring positions have a certain influence on the motion state of the measuring ruler. As Figure 8 shows, six measuring positions were selected on the measuring ruler as *a* to *f*, and the distances from the left reference point *o* were 10 cm, 30 cm, 40 cm, 50 cm, 60 cm, and 130 cm, respectively. The vehicle speed was set to 15 km/h during simulation and the motion process of the measuring ruler was simulated under different measuring positions, as shown in Figure 9. As seen in this Figure, the most violent vibration positions were *a* and *f*, which were at both ends of the measuring ruler. The best measuring position was *d*, which was almost not affected by vibration.

#### 4.2.3. Selection of the Downward Displacement of the Measuring Ruler

The downward displacement of the measuring ruler under wheel pressure could be set by adjusting the position of the limiting blocks. The simulation results shown in Figure 10 indicate that reducing the downward displacements can alleviate the impact vibration. When the train speed was 30 km/h, no vibration occurred on the measuring ruler with displacement below 2 mm, as shown in Figure 10a. When the displacement was 1 mm, the velocity threshold could rise to 40 km/h, as Figure 10b shows.

## 5. Model Validation

Figure 11 shows that the whole system mainly consists of four parts: the parallelogram mechanisms, the displacement transducer system, the data collection and processing system, and the computer and communication system. To detect tread flats on the entire circumference of the wheel, and avoid two wheels of the same bogie pressing on the same parallelogram mechanism, the following two conditions were met:
Three sets of parallelogram mechanisms and the corresponding eddy current sensors were installed on the outside of each rail, and the sum of the measuring ruler lengths of the three sets of parallelogram mechanisms should be greater than the circumference of the wheel.The measuring ruler length of each parallelogram mechanism must be less than the minimum wheelbase.

In this system, the length of the measuring ruler was 1400 mm. When the train passed through the measuring mechanism, the detection signal generated by the eddy current sensor was pre-amplified and processed before being input into the computer. The computer processed the measured waveform of the measured wheel. When the wheel flat signal appeared in the measured waveform, the flat wheel could be located by analyzing the time sequence of the whole train waveform.

### 5.1. Laboratory Experiments

To test system performance, laboratory experiments were carried. The wheel passed through the system at a low speed of 5 km/h. In the measuring mechanism, there are eight gears for the hydraulic damper adjustment, and three gears (gear 0, gear 6, and gear 8) were adjusted in the experiment. The higher the gears, the larger the damping coefficients. Measurements under different gears were conducted and the results are shown in Figure 12. As Figure 12 shows, the variation trend of the curves is consistent with the simulation results in Section 4.2.1. According to the experimental and simulation results, gear 0 of the hydraulic damper was selected due to its good performance.

Gaskets with different thickness (0.1 mm, 0.15 mm, 0.2 mm, and 0.3 mm) were used to simulate the wheel flat. The gaskets were placed on the rail with the parallelogram mechanism. The measured waveform obtained after the wheel passes is shown in Figure 13. Since the wheel rolls through the raised gasket with an upward displacement, the measured sudden change signal is upward, which is opposite to the actual wheel flat signal direction. The measured results shown in Table 1 demonstrate that the impact signal of the parallelogram mechanism can be accurately obtained with an error of ±0.05 mm.

To verify the effectiveness of the system, a flat wheel was measured in the lab at a low speed of 5 km/h. The measured waveform of the flat wheel is shown in Figure 14. The position of the tread flat caused a sudden change in the measurement the signal, and the amplitude of the signal reflected the wheel flat depth. The calculation showed that the wheel flat depth was 0.3 mm, and the flat length was 32 mm.

### 5.2. Field Experiments

To verify the model and simulation results, field tests were performed at the Baolan railway in Yinchuan, China during April 2019. The system was installed at the entrance of the train garage as shown in Figure 15.

There was no theoretical basis for the installation location and the system parameter selection before the simulation, which resulted in poor system performance. The measurement results under different train speeds before parameter reasonable selection is shown in Figure 16, and we found that system performance was influenced by the speed of the train. When the train speed increased from 14 km/h to 15 km/h, the vibration amplitude of the measured waveform rapidly increased. The results were consistent with the simulation results in Section 4.1.2.

Then the system parameters were selected reasonably based on the simulation results. In the experiments, the eddy current displacement sensor was installed at the optimal measurement position obtained using the simulation in Section 4.2.2, where the limit block was adjusted to allow the downward displacement of the measuring ruler at 0.5 mm. Since the measuring mechanisms were installed at the entrance of the train garage, the maximum speed of the train entering the train garage was 15 km/h, which is limited by the local railway management. The measured waveform with the maximum passing speed of 15 km/h was validated, as shown in Figure 17. In Figure 17, before the selection of the mechanism parameters, the vibration of the measuring ruler was obvious and the maximum amplitude was 0.75 mm, and after reasonable selection, the vibration amplitude was reduced to 0.25 mm. Therefore, after reasonable selection of the parameters, the vibration could be reduced and the speed threshold of the measuring ruler could be improved.

## 6. Conclusions

A parallelogram mechanism based on the wheel flat detection system was demonstrated to realize the dynamic and quantitative measurement of wheel flats. To improve the speed threshold of the measuring mechanism vibration under wheel impact, we established a three-dimensional simulation model based on the rigid-flexible coupled multibody dynamics theory with the parallelogram mechanism. The selected parameters included the damping coefficients of a hydraulic damper, the measuring positions, and the downward displacements of the measuring ruler. Results of experiments performed in the laboratory demonstrated the effectiveness of the system. Results of experiments performed in the field demonstrated that the velocity threshold of the measuring ruler increased after reasonably selecting the model parameters that met requirements in the field. It verified that the measured waveform at a train at speed 15 km/h reduced the vibration by 0.5 mm compared to parameters before reasonable selection, which proved the correctness of the model simulation.

## Figures and Tables

**Figure 1 sensors-19-03614-f001:**
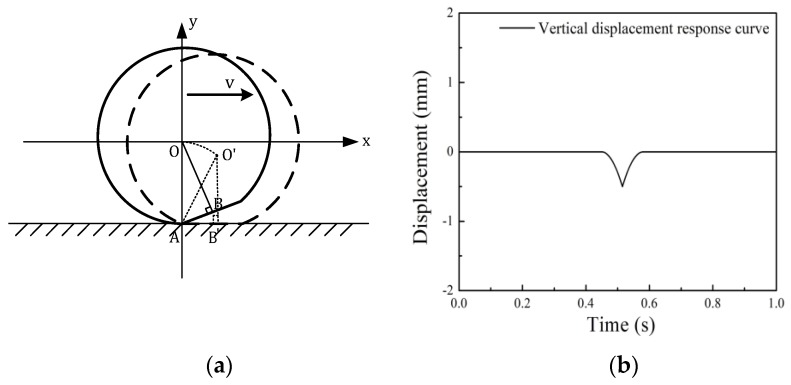
The vertical motion of a wheel flat: (**a**) The geometry of a wheel flat; (**b**) The vertical displacement response curve of point *O*.

**Figure 2 sensors-19-03614-f002:**
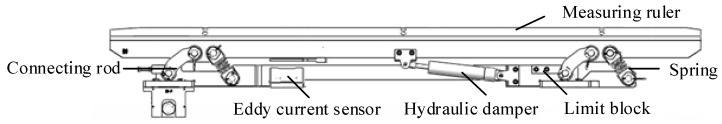
A schematic diagram of the parallelogram mechanism.

**Figure 3 sensors-19-03614-f003:**
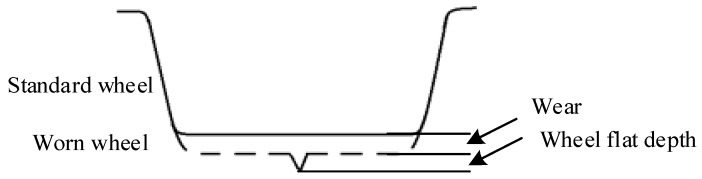
An output waveform of the measuring device under ideal conditions.

**Figure 4 sensors-19-03614-f004:**
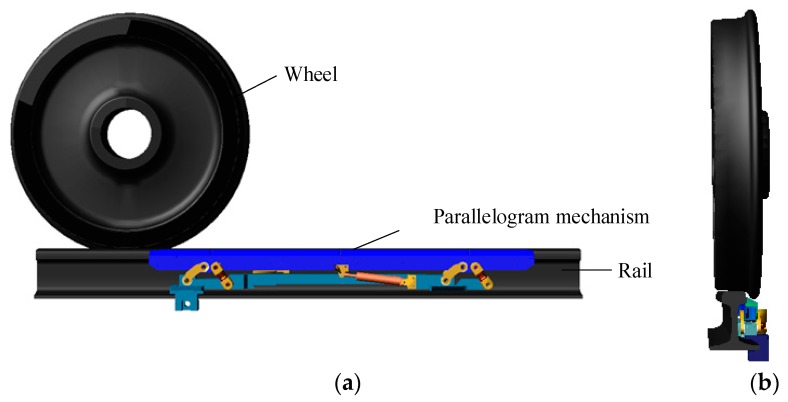
The dynamic simulation model: (**a**) The front view; (**b**) The side view.

**Figure 5 sensors-19-03614-f005:**
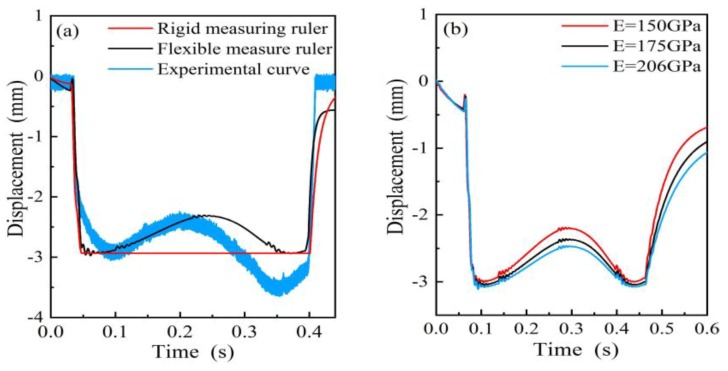
The measuring ruler motion waveform: (**a**) The simulation curve and the experimental curve; (**b**) The motion curve with different Young’s modulus.

**Figure 6 sensors-19-03614-f006:**
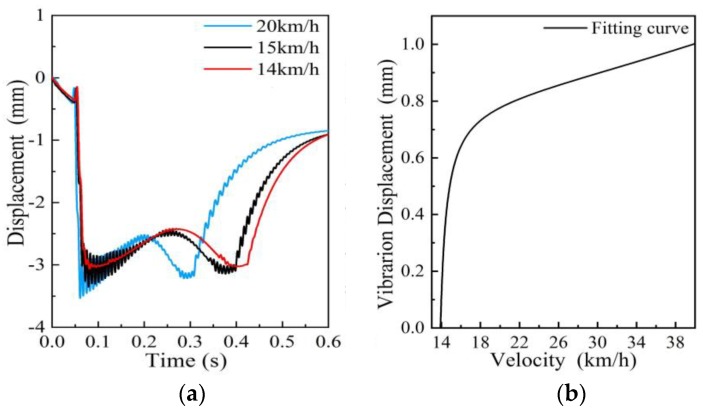
The influence of velocity on the motion process of the measuring ruler: (**a**) Motion curve with different speeds; (**b**) Relationship between speeds and maximum vibration displacements.

**Figure 7 sensors-19-03614-f007:**
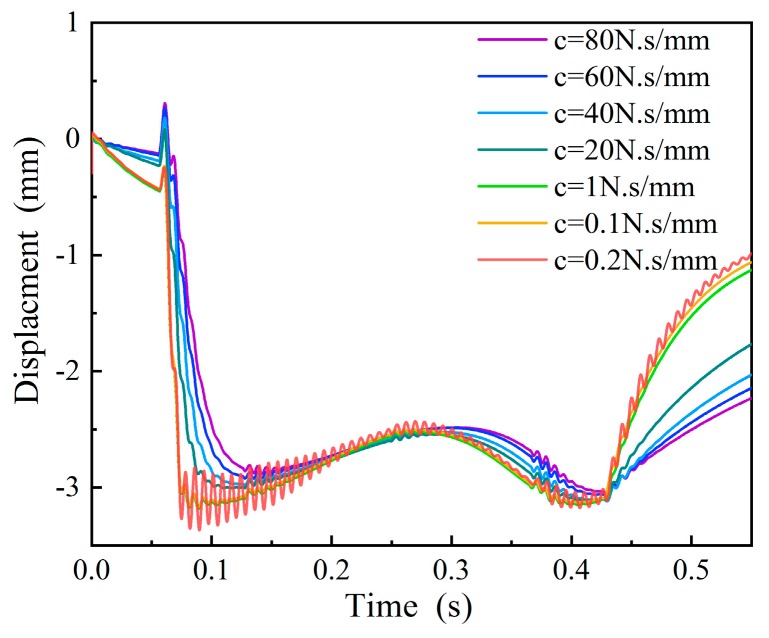
The motion states of the measuring ruler with different damping coefficients.

**Figure 8 sensors-19-03614-f008:**
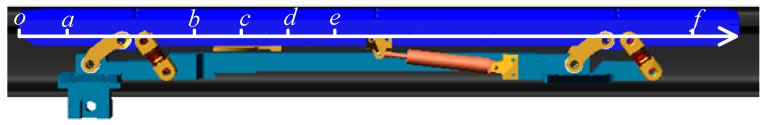
The schematic diagram of different measuring positions at the measuring ruler.

**Figure 9 sensors-19-03614-f009:**
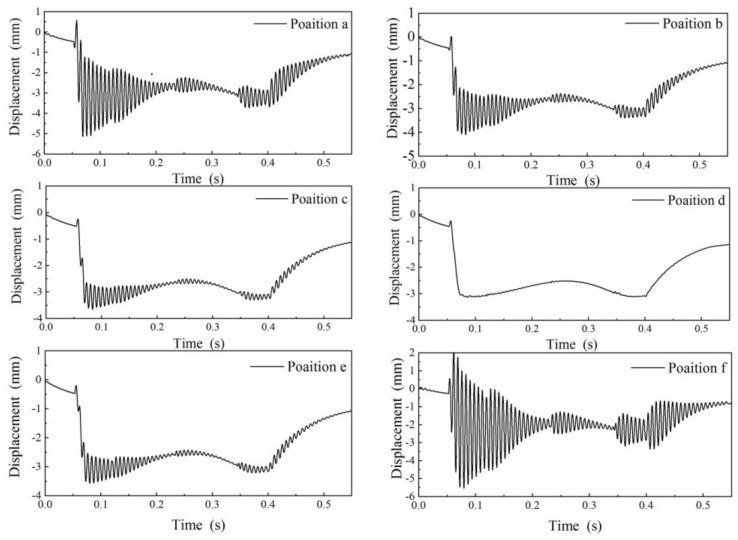
The motion state of the measuring ruler at different measuring positions.

**Figure 10 sensors-19-03614-f010:**
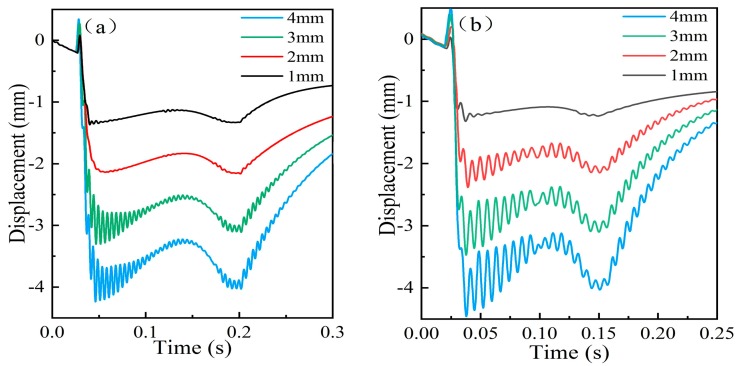
The motion state of the measuring ruler with different downward displacement: (**a**) The motion curves of the measuring ruler with different displacements at 30 km/h; (**b**) The motion curves of the measuring ruler with different displacements at 40 km/h.

**Figure 11 sensors-19-03614-f011:**
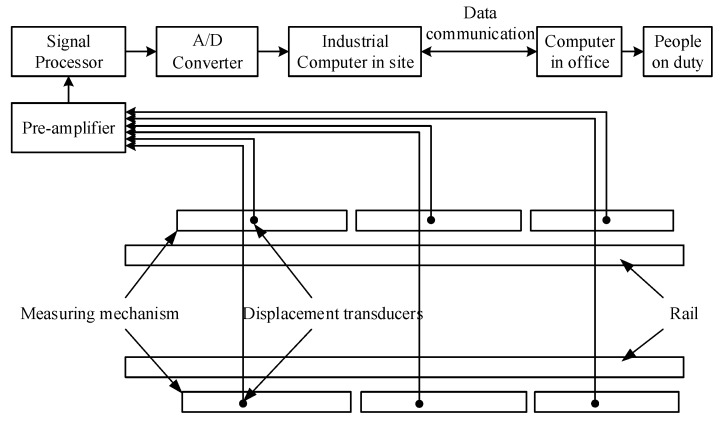
The architecture of the wheel flat detection system.

**Figure 12 sensors-19-03614-f012:**
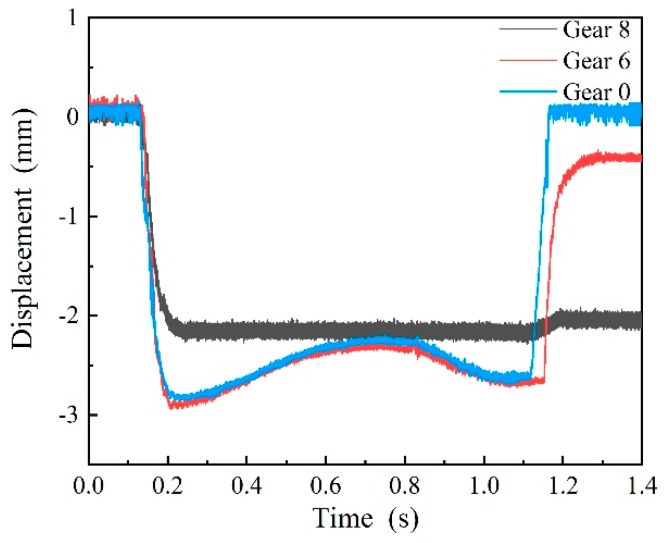
The motion process of the measuring ruler at different hydraulic damper gears on site.

**Figure 13 sensors-19-03614-f013:**
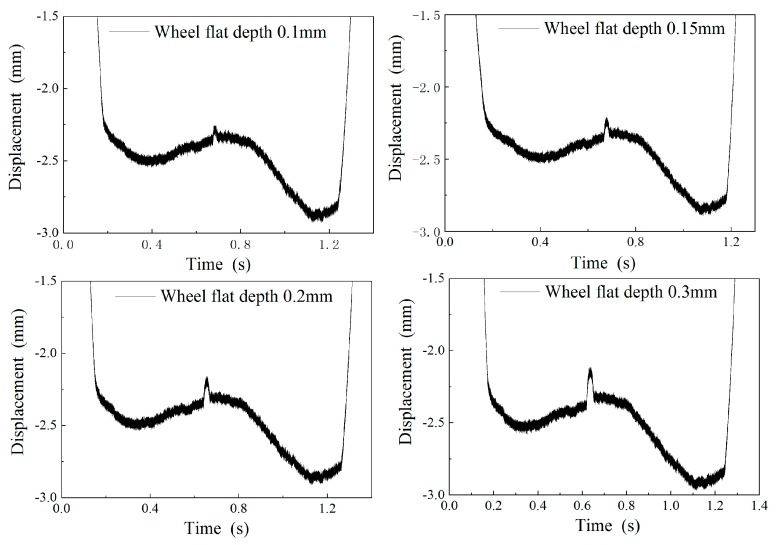
The measuring waveform of the simulated wheel flat.

**Figure 14 sensors-19-03614-f014:**
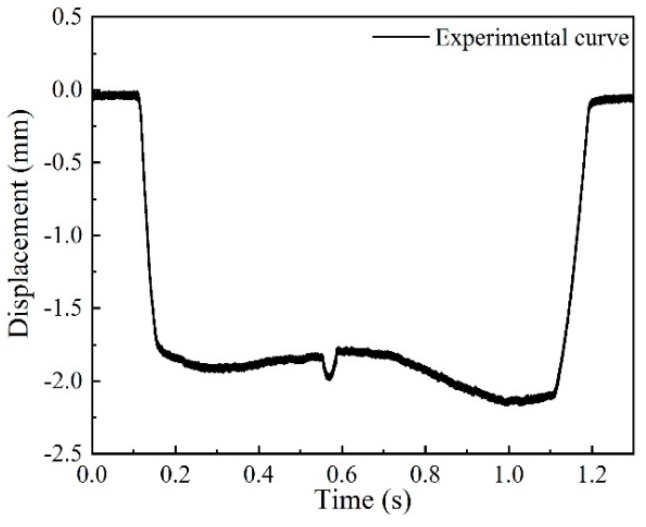
The measuring waveform of a wheel flat.

**Figure 15 sensors-19-03614-f015:**
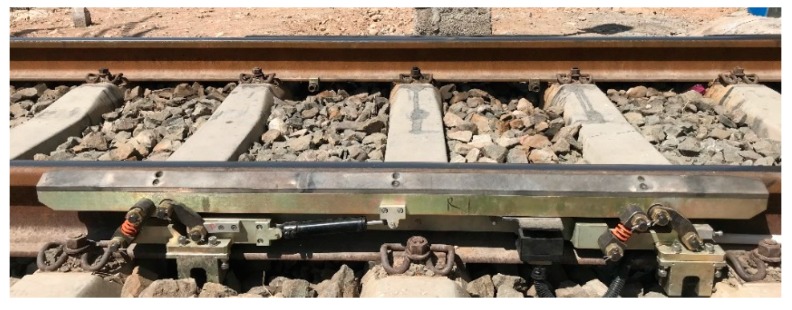
The photo of the installed parallelogram mechanism.

**Figure 16 sensors-19-03614-f016:**
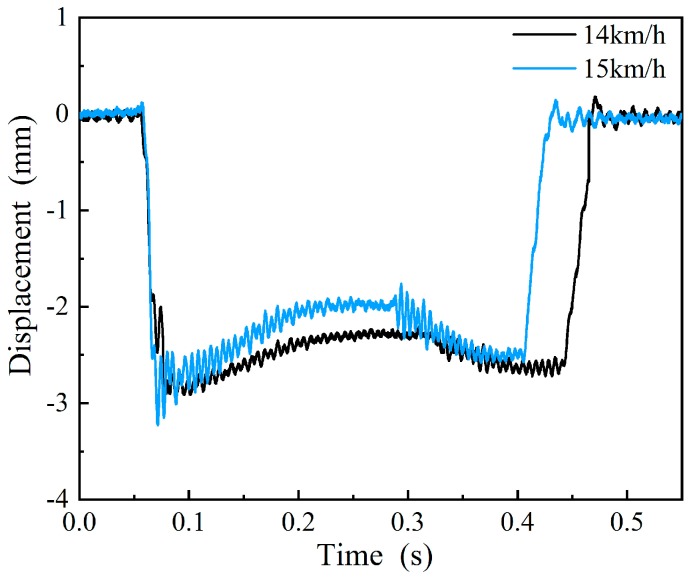
The motion process of the measuring ruler at different vehicle speeds on site.

**Figure 17 sensors-19-03614-f017:**
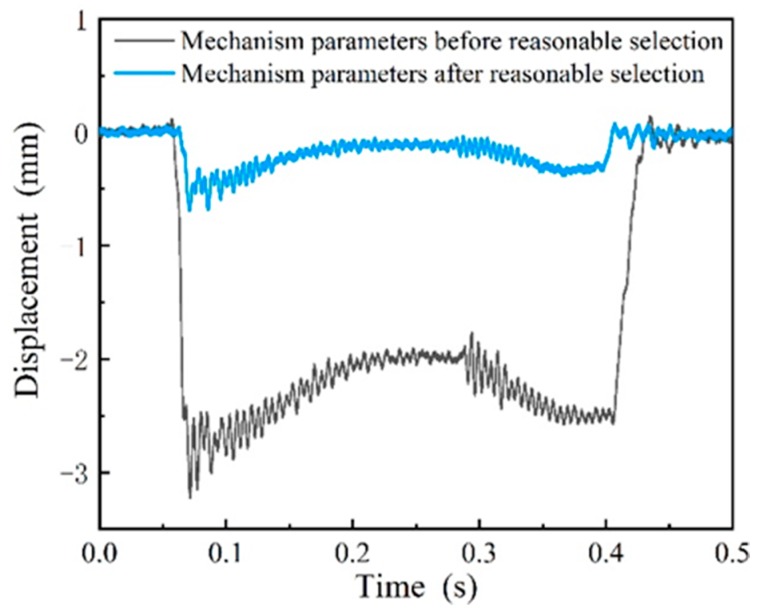
The motion process of the mechanism parameters before and after reasonable selection.

**Table 1 sensors-19-03614-t001:** Wheel flat measurement results.

Parameter	Wheel Flat Depth			
Theoretical value [mm]	0.10	0.15	0.20	0.30
Measured value [mm]	0.15	0.17	0.21	0.29
Measured Error [mm]	0.05	0.02	0.01	0.01

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
