# Peer review of "Railway Wheel Flat Detection System Based on a Parallelogram Mechanism"

_sensors, 2019, doi:10.3390/s19163614_

Round 1
Reviewer 1 Report
The manuscript presents a method for the dynamic and quantitative measurement of wheel flats which is an important and valuable work. The wheel flat detection system was built and the performance was optimized by a three-dimensional simulation model. The model was verified by field test and can present theoretical evidence for the system installation and selection principle of components. Based upon this, I consider the work is suitable for publications.
However, before publication, there are a few instances at which I think that the paper is not clear enough that need to be taken care of. These are itemized below.
1. Please describe the measuring ruler in Section 1.
2. Parts of the text in Section 2.2 is written in future tense: Eg “the parallelogram mechanism will be pressured down…”. This is unsuitable and should be changed.
3. It was written: 'there are 8 gears for the hydraulic damper adjustment, and three gears (gear 0, gear 6 and gear 8) were adjusted in the experiment' in Section 5.1, why choose these three gears?
4. It was written: 'Then, the signal processing system can locate the abrasion wheel within a few seconds' in Section 5. This should be clear.
5. In Table 1, the measured error decreases with the increase of wheel flat depth, please explain it.
6. The authors should include some more recent papers published in Sensors and other journals that discuss the detection of wheel flats.
Reviewer 2 Report
please see the doc. attached

Reviewer 3 Report
The reviewed paper aims at very interesting topic - detection of wheel flat. The wheel flat as well as other wheel unevenness and defects represent phenomenon, which could case serious accidents in railway transport. Therefore, there is necessary to detect them early. Authors present interesting study and design of such a device, which is able to to it.
Contents of the article consists of theoretical background, which the practical application of the designed device comes from. Authors present principle of a device, a method of its application to a rail track and a method of evaluation.
In my oppinion, it is worth to continue in this field of research.

Reviewer 4 Report
In paper under review authors described railway wheel flat detection system based on parallelogram mechanism.
In the first chapter, the authors reviewed wheel flat detection methods. More than 75% of the literature is older than five years.
Line 27. Authors use nine [1-9] references for one simple sentence, same situation - line 33. In literature, there is no enough motivation, which is the main problem in wheel flat detection, which is still not solved. Why the parallelogram mechanism is needed.
Line 70. “The rotation speed of the point O is ω=v/R and the motion equation of point O in the vertical is as follows”. In reality, the angular velocity in point O (wheel centre) is equal to 0, as R=0 in that point. The sentence should be reviewed.
Section 3. Grid-flex multibody dynamics model.
Only general equations are presented without any details for the case under investigation. Also, a wheel with flat-parallel quadrilateral mechanism is the nonlinear system. For the nonlinear system, the superposition method can’t be applied, so the additional description is required for clarification.
Line 100. “The dynamic equation of measuring ruler…” maybe the equation of motion can be used instead.
Line 110. “the frequency of the j modal shape, ?? = sqrt(??/?n)” this equation is used for 1 degree of freedom systems, for the system under investigation: ([K]-w^2[M])=0 are used. In both this equations damping is not taken into account, as authors calculate damping, natural frequencies can be calculated with damping.
In section 3.2, the wheel is taken into account, but it wasn't taken into account in equations (3) –(11).
Chapter 3 should be revised, as there is a lot of questions, why it is needed.
Chapter 4, Parameter optimisation.
In this chapter, there no optimisation. Authors solve a task using different parameters, but the optimisation task is not solved. May be chapter can be renamed to parameter selection.
Line 146. “Where ? represents the material stiffness…” according to equation 12 it is contact stiffness between bodies. What is STEP in eq. 12?
Round 2
Reviewer 2 Report
I am satisfied with the author's reply on my previous comments and the manuscript revision, so, I agree with its acceptation for publication.
Reviewer 4 Report
Thank you for corrections, article is improved.
Heaviside function can be used instead of STEP in equation 3, but STEP is also acceptable.